# Progerin, an Aberrant Spliced Form of Lamin A, Is a Potential Therapeutic Target for HGPS

**DOI:** 10.3390/cells12182299

**Published:** 2023-09-18

**Authors:** Bae-Hoon Kim, Yeon-Ho Chung, Tae-Gyun Woo, So-Mi Kang, Soyoung Park, Bum-Joon Park

**Affiliations:** 1Rare Disease R&D Center, PRG S&T Co., Ltd., Busan 46274, Republic of Korea; bk728@prgst.com (B.-H.K.); yhchung94@prgst.com (Y.-H.C.); taegyun0728@prgst.com (T.-G.W.); 2Department of Molecular Biology, College of Natural Science, Pusan National University, Busan 46231, Republic of Korea; rosa.k0526@pusan.ac.kr (S.-M.K.); thdud2971@naver.com (S.P.)

**Keywords:** Hutchinson–Gilford progeria syndrome, progerin, nuclear lamina

## Abstract

Hutchinson–Gilford progeria syndrome (HGPS) is an extremely rare genetic disorder caused by the mutant protein progerin, which is expressed by the abnormal splicing of the *LMNA* gene. HGPS affects systemic levels, with the exception of cognition or brain development, in children, showing that cellular aging can occur in the short term. Studying progeria could be useful in unraveling the causes of human aging (as well as fatal age-related disorders). Elucidating the clear cause of HGPS or the development of a therapeutic medicine could improve the quality of life and extend the survival of patients. This review aimed to (i) briefly describe how progerin was discovered as the causative agent of HGPS, (ii) elucidate the puzzling observation of the absence of primary neurological disease in HGPS, (iii) present several studies showing the deleterious effects of progerin and the beneficial effects of its inhibition, and (iv) summarize research to develop a therapy for HGPS and introduce clinical trials for its treatment.

## 1. Introduction

Hutchinson–Gilford progeria syndrome (HGPS; OMIM#176670) was first reported more than 100 years ago by Hutchinson and Gilford in 1886 and 1897, respectively [1]. The condition was designated as a premature aging syndrome by Gilford based on the fact that the symptoms associated with aging are similar to the changes seen in older people in general, including a lack of subcutaneous fat, hair loss, joint contractures, a progressive cardiovascular disease similar to atherosclerosis, and death from heart attacks and strokes in childhood. As patients typically live to their teens or early 20s and usually die before reaching reproductive age, this syndrome is not inherited. Diagnosis can be achieved within the first 6 months of age, although prominent and noticeable symptoms may be observed later [2,3,4,5,6,7,8]. This fatal pediatric disease remained a medical mystery until genetic mapping revealed that 90% of patients have a de novo point mutation in the *LMNA* gene that replaces cytosine with thymine [9,10].

Nuclear membrane proteins (lamins A and C), encoded by the *LMNA* gene, are structural components of the nuclear lamina, a network of proteins underneath the nuclear membrane that determines the shape and size of the nucleus [11,12]. *LMNA* produces four proteins as a result of alternative splicing: lamin A and lamin C as major products, and lamin C2 and lamin A delta 10 as minor products. Lamins A and C are similar through the first 566 amino acids (encoded by exons 1–10) but deviate at the carboxyl terminus [3,13,14]. Prelamin A, but not lamin C, contains a CaaX motif at its C-terminus and undergoes farnesylation and methylation (Figure 1). Lamin A is synthesized as a precursor (prelamin A) and matures through four sequential post-translational processing steps [15]. First, farnesyltransferase (FTase) adds a 15-carbon farnesyl moiety to the carboxyl-terminal cysteine. Second, the Zmpste24 endopeptidase cleaves the last three amino acids of prelamin A. Third, the newly exposed farnesyl cysteine is carboxyl-methylated by a prenyl protein-specific methyltransferase. Finally, the endopeptidase removes 15 carboxyl-terminal amino acids from the protein, resulting in the release of mature lamin A. 

It has been questioned whether the post-translational processing steps of prelamin A are essential in targeting the protein to the nuclear envelope [16]. Mice that could directly produce mature lamin A without going through the usual prelamin A synthesis and processing steps were created. However, no detectable disease phenotype was observed in the mice and the nuclear membrane of mature lamin A appeared normal [16], suggesting that prelamin A processing is minimally important for the nuclear targeting of mature lamin A and is independent of lamin B in laboratory mice.

HGPS belongs to a group of diseases called laminopathies, in which mutations across the *LMNA* gene result in a wide range of overlapping disorders [17]. Genetic mapping of the genome from patients elucidated that a sporadic, autosomal-dominant de novo point mutation, c.1824C>T (p.G608G) (NM_170707.3) in exon 11 of the human *LMNA* gene, mediates abnormal alternative splicing [9,10], which produces an abnormal variant protein called progerin, which is responsible for this accelerated aging disease [18,19,20] (Figure 1).

The current review describes the discovery of progerin as a causative agent of HGPS and provides evidence of its deleterious effects when expressed intracellularly, and the benefits of inhibiting its expression. Additionally, it introduces efforts to develop therapies and clinical trials for HGPS.

## 2. Absence of Primary Neurological Disease in HGPS 

Extensive studies have been conducted to examine mutations in the *LMNA* gene encoding prelamin A and lamin C, which result in distinct muscular dystrophy, cardiomyopathy, partial lipodystrophy, and progeroid syndromes. These laminopathies mostly affect mesenchymal tissues (e.g., the myocardium, skeletal muscle, adipose tissue, fibrous connective tissue, and bone tissues). However, one confusing observation in patients with HGPS is that they generally show fundamental and dramatic premature aging but do not exhibit any noticeable cognitive damage. For many years, it has been puzzling that patients with HGPS do not have any primary neurological disease. However, recent research has confirmed a lack of lamin A expression, the major isoform of *LMNA*, in HGPS patient-driven induced pluripotent stem cells (iPSCs) [21,22,23,24]. In most tissues, the amounts of lamins A and C are approximately equal; however, the brain mostly generates lamin C and very little lamin A [21,25]. Immunohistochemistry has indicated that lamin C is expressed at high levels in the neurons and glia of the brain, but the expression of prelamin A and lamin A is restricted to the vascular endothelial cells [25]. Further studies have shown that the expression of prelamin A in the brain is downregulated by miR-9, a microRNA highly expressed in the brain that binds to a single site in the 3′ untranslated region of prelamin A [21,22,25,26]. The ectopic expression of miR-9 in fibroblasts or HeLa cells decreases the levels of prelamin A and lamin A proteins but does not influence lamin C expression [25]. In lamin-A-only knock-in mice, where there is no alternative splicing and the output of all genes is directed to the prelamin A transcript, high levels of lamin A are found in the peripheral tissues, but very little lamin A is found in the brain [25]. Likewise, a knock-in mouse was created to direct the production of LMNA towards the progerin transcript. In this model, high levels of progerin were expressed in peripheral tissues, while minimal levels were observed in the brain [25], demonstrating that the unique expression pattern of lamin A/lamin C in the brain is not the result of alternative splicing. This regulation of lamin A in the brain provides us with a basis to further study improperly processed progerin and toxic lamin A. Children with HGPS have aging-like phenotypes in many tissues but lack common features of physiological aging in the central nervous system (CNS), such as senile dementia. Therefore, progerin accumulation in cells is considered a pathology-inducing factor.

Additionally, several studies have highlighted the important role of nuclear lamins in the CNS, indicating that type B lamins, lamins B1/B2, play an important role in neuronal migration in the developing brain [27,28,29]. Duplication of the *LMNB*1 gene encoding lamin B1 has been shown to cause autosomal-dominant leukodystrophy (ADLD) [30,31]. More recently, both *LMNB*1 deficiency and overexpression have been reported to inhibit proliferation, but only *LMNB*1 overexpression induces senescence, which is prevented by telomerase expression or p53 inactivation. A concomitant decrease in lamin A/C levels aggravates this phenotype. These findings show that changes in the expression of *LMNB*1 inhibit proliferation and are potentially relevant in understanding the molecular pathophysiology of ADLD [32], suggesting the possibility that a distinct spectrum of “brain laminopathies” might eventually be mapped to missense mutations in *LMNB*, not in *LMNA*.

## 3. Dysfunctional Progerin Expression

Effect of progerin at the cellular level

Notably, the expression levels of progerin and lamins A and C (lamin A/C) were significantly reduced in iPSCs derived from patients with HGPS [21,22,23,24]. Moreover, these cells showed decreased patterns of cellular senescence markers, including nuclear deformation, histone H3 trimethyl Lys9 (H3K9-Me3), and senescence-associated β-gal (SA–β-gal). In contrast, HGPS cells differentiated from iPSCs start expressing progerin and lamin A/C and re-expressing senescence markers [23]. This implies that the expression of the *LMNA* gene is tightly regulated at an early developmental stage; therefore, progerin is expressed in differentiated HGPS cells and its expression drives the cells to a pathological state. Twenty years ago, Collins et al. proved that progerin is the main factor inducing a premature aging phenotype in HGPS by the disruption of lamin-related functions ranging from the maintenance of nuclear shape to the regulation of gene expression and DNA replication [18]. 

A correlation between progerin levels and the severity of HGPS phenotypes has been reported. Progerin levels in HGPS fibroblasts increase with the culture passage number [19,33,34]. Recently, Gordon et al. developed a plasma assay to assess the amount of progerin in response to progerin-targeted therapy and its correlation with patient survival. The extent of the survival improvement was related to both the magnitude and duration of progerin reduction at low levels, demonstrating that the level of plasma progerin is a biomarker of HGPS that enables the short- and long-term assessment of progerin-targeted therapeutic efficacy through progerin reduction [35].

The arrangement of chromatin in the nucleus is crucial in regulating many aspects of nuclear function and protecting nuclear integrity. The nuclear envelope (NE) is an essential factor in the dimensional scattering of these chromosomes. It binds to and ties a wide range of chromatin domains by interacting with the nuclear lamina and other associated proteins. Progerin sequesters NRF2 at the NE site, causing a subnuclear localization mismatch that impairs NRF2 transcriptional activity and consequently increases chronic oxidative stress [36]. Progerin also induces altered 3D telomere organization between telomeres and the nuclear lamina, and an altered telomeric chromatin state [37]. Overexpression of telomerase reverse transcriptase (TERT) enhances the proliferative capability of HGPS fibroblasts and repairs progerin-induced DNA damage [38,39]. Chojnowski et al. [40] suggested a controllable cellular model of progeria and showed that exogenous TERT prevents proliferation deficiency, DNA damage, lamin B1 reduction, and gene expression differences induced by progerin, suggesting that progerin disturbs the interaction between LAP2α and telomeres. However, TERT could not restore defects in nuclear morphology or altered H3K27me3 deposition [40]. Other reports have indicated that progerin expression is sufficient to cause heterochromatin loss by inducing DNA damage [41,42].

ER stress can be induced by the disruption of cellular energy levels, such as the redox status or Ca^2+^ concentration, leading to the accumulation and aggregation of unfolded proteins [43,44]. Progerin expression disrupts calcium homeostasis and whole-body energy consumption. A mouse model (CAG-Progerin+ and MCK-Cre+) was developed to investigate the role of human progerin in sarcoplasmic reticulum function [45]. The study demonstrated that the conditional overexpression of progerin in muscle tissue is sufficient to provoke premature death and impair the regulatory control of the expression of thermogenesis-related genes. 

It has been found that the abnormal expression of progerin could aggravate the defenestration of liver sinusoidal endothelial cells (LSEC) during liver fibrosis, whereas the knockdown of progerin expression could attenuate premature liver senescence [46,47]. It has been proposed that in response to DNA damage, the binding between LC3 and UBC9-sumoylated lamin A/C enables autophagy to specifically mediate the disruption of lamins A/C and leaky nuclear DNA, suggesting that, in response to cancer-promoting stresses such as DNA damage, autophagy breaks down nuclear material to prevent tumor formation [48]. Another study has shown that progerin is involved in nucleophagy [49]. It was indicated that deficient nucleophagy due to progerin expression caused oxidative stress presumably due to the decomposition of basal lamin B1. Furthermore, the acetylation of nuclear LC3 was responsible for the unusual deposition of progerin, whereas its deacetylation promoted progerin removal, thus suggesting a potentially novel approach to maintaining the LSEC phenotype [50].

Progerin-mediated functions at the protein level and their relationship with miRNA expression have also been described in recent studies [51,52,53]. A structure-based study has revealed that unfarnesylated progerin can form a disulfide bond with an Ig-like domain in the nuclear lamina. The Alphafold2-assisted docking structure showed that disulfide bond formation was promoted by a weak interaction between the groove of the Ig-like domain and the unfarnesylated C-terminal tail region of progerin, providing molecular insights into the abnormal interactions caused by progerin [51]. The evaluation of miRNA expression profiles in HGPS and normal fibroblasts revealed an enriched set of overexpressed miRNAs belonging to the 14q32.2–14q32.3 miRNA cluster and showed that inducing their overexpression in normal fibroblasts reduced cell proliferation and increased senescence, whereas inhibiting them in HGPS fibroblasts alleviated proliferation defects and senescence and reduced progerin accumulation [52]. Progerin overexpression induced notable changes in miRNA expression and confirmed that has-miR-59 (miR-59) was markedly upregulated in cells from patients with HGPS and in multiple tissues of an HGPS mouse model (*Lmna*^G609G/G609G^) [53]. 

In this report, we describe the relationship between progerin and cellular homeostasis, and its role in inducing premature aging in most cells. Several groups have examined progerin as a biomarker of aging and indicated that it may be one of the few known biological indicators that initiate the aging process at a certain age [54,55,56,57]. Recently, fibroblasts cultured from older individuals were shown to have nuclear features similar to HGPS cells [54]. Notably, although these cells expressed progerin mRNA transcripts at barely detectable levels [54], they contained several abnormal nuclei that were clearly positive for progerin-specific antibodies after prolonged culture [58,59], indicating that progerin can be expressed in normal cells. To further investigate the biological relationship between progerin expression and aging in normal humans, Djabali et al. examined skin biopsies from 150 unaffected individuals. They found that similar splicing events occurred in vivo at low levels in the skin of individuals of all ages [60]. Although the mRNA expression level of progerin is low, it accumulates with age in a subpopulation of skin fibroblasts and terminally differentiated keratinocytes [60,61], suggesting that research on HGPS may improve our knowledge of physiological aging. 

b.Systemic effect of progerin or effect of progerin on tissues

Over the last 20 years, several types of mice have been developed as animal models to investigate diverse aspects of HGPS as follows: the knock-out or transgenic mice affecting the whole-body level are *Zmpste24^−/−^* [62,63], transgenic G608G BAC [64], *Lmna*^G609G^ [65], *Apo^−/−^ Lmna*^G609G/G609G^ [66], *Ldlr^−/−^,* and *Lmna*^G609G/G609G^ [67]; the mice affecting specific tissues or cells are *Lmna*^LCS/LCS^
*SM22αCre* [66], *Apoe^−/−^ Lmna*^LCS/LCS^
*SM22αCre* [66], *LmnaLCS/LCS Tie2Cre* [68,69], *Lmna*^f/f^; *TC* [70], and *Prog-Tg* [71]—see reference [72] for further details. The limited number of patients with HGPS worldwide renders it difficult to conduct longitudinal studies and clinical trials, and there is also a scarcity of human samples available for ex vivo analyses. Therefore, the use of animal models and their derived cell lines has contributed to the understanding of progeria phenotypes and is particularly important in developing therapeutic reagents for the disease. The current section aims to examine how progerin affects the phenotypes of various cell types and mouse models by compiling research carried out by various experts. 

Blood vascular diseases are the predominant cause of death in classical HGPS [73]. As childhood progresses, other symptoms appear, including hair loss, joint stiffness, body fat loss, osteoporosis, and other aspects of physiological aging. The most clinically relevant aspect of HGPS is the hardening of the arteries (atherosclerosis), which leads to premature death from cardiovascular disease or stroke at an average age of approximately 15 years [74]. In HGPS, atherosclerosis is accompanied by pathological changes in the aortic wall, including severe vascular smooth muscle cell (VSMC) depletion in the media, extracellular matrix deposition, calcification, and early thickening of the aortic wall [61,75,76]. Skin tissue sections from patients with HGPS have shown that progerin accumulates primarily in the nuclei of vascular cells, suggesting that its accumulation has a direct association with vascular diseases in progeria [19]. Similarly, several reports have suggested that progerin in vascular muscles could accelerate atherosclerosis by inducing endoplasmic reticulum (ER) stress, DNA damage, wound healing impairment, mislocalization of a myocardin-related transcription factor, and replication stress [77,78,79,80,81]. Recently, Hamczyk et al. (2018) produced the first mouse model (*Apoe*^–/–^*Lmna^LCS/LCS^SM22αCre*) with progerin-induced atherosclerosis acceleration expressing progerin specifically in VSMCs and demonstrated that restricting progerin expression to VSMCs is sufficient to accelerate atherosclerosis, trigger plaque vulnerability, and reduce lifespan [66]. Moreover, they succeeded in developing CRISPR-Cas9 technology to generate HGPSrev mice (*Lmna^HGPSrev/HGPSrev^*), engineered to express progerin throughout the body while lacking lamin A and allowing progerin suppression and the restoration of lamin A in a temporal and cell-type-specific manner upon Cre recombinase activation. Regardless of the broad expression of progerin and its pathological effects in several organs, restricting its suppression to VSMCs and cardiomyocytes is adequate to ameliorate vascular diseases and extend the lifespan of mouse models [82].

Progerin accumulation is associated with fat tissue disorders [83,84,85] and its expression decreases the capacity for adipocyte differentiation in both iPSCs and human mesenchymal stem cells (hMSCs) derived from patients with HGPS [57,86]. Mateos et al. [87] performed quantitative proteomics to study the effect of progerin accumulation in a preadipocyte cell line, 3T3L1 cells. They reported that progerin accumulation in adipocytes contributed to the generation of reactive oxygen species and premature aging features, establishing a relationship between mitochondrial malfunction and proteostasis failure in HGPS [87]. 

Patients with HGPS exhibit unique skeletal dysplasia with bone morphological abnormalities and short stature. The HGPS mouse model (homozygous transgenic G608G BAC) also shows a similar bone structure pattern [88,89,90]. The cartilage abnormalities observed in this HGPS mouse model were similar to those observed in age-matched WT controls, including the premature loss of glycosaminoglycans and decreased cartilage thickness and volume. These alterations may mimic degenerative joint diseases prevalent in the elderly [89]. The Zmpste24^−/−^ HGPS and progeria mouse model showed the development of kyphosis and spontaneous bone fractures in multiple locations [91]. More recently, the *Lmna*^G609G/G609G^ mouse model exhibited joint immobility and skeletal deformities in the vertebral column and skull [90].

Several studies have examined the relationship between progerin and inflammatory responses [92,93]. Endothelial cells expressing progerin recapitulate some characteristics of aging-associated cell dysfunction, including pro-inflammatory features, oxidative stress, DNA damage, increased expression of cell cycle arrest proteins, and cellular senescence [92]. Using HGPS fibroblasts, it was shown that progerin-induced replication stress causes genomic instability by stalling the replication fork and nuclease-mediated degradation, along with the upregulation of the cGAS/STING cytosolic DNA sensing pathway and the activation of a robust STAT1-regulated interferon (IFN)-like inflammatory response [93]. Hamczyk et al. (2018) also showed that exogenously expressed progerin increased inflammation [66]. A significant correlation was observed between chronic inflammation and ZMPSTE24 levels. Additionally, patients with cardiovascular diseases showed abnormal lamin A/C expression associated with progerin levels [94]. The pro-atherogenic role of progerin in HGPS-related early atherosclerosis was proposed by Bidault et al. [92]. They found that progerin overexpression increased the expression of pro-inflammatory cytokines IL-6 and IL-1β, intercellular adhesion molecule-1 (ICAM-1), and vascular cell adhesion molecule-1 (VCAM-1), enhancing inflammation along with oxidative stress [92]. González-Dominguez et al. (2021) recently suggested that progerin was responsible for the activation of the NLRP3 inflammasome [95], which is a multiprotein complex having an intracellular sensor comprising NLRP3 itself, the adapter protein ASC, and the catalytic subunit (Caspase 1), which cleaves pro IL-1β to the mature IL-1β. Furthermore, it was revealed that the activation was associated with alterations in nuclear morphology, indicating its relation to the induction of IL-1β by progerin [92,95]. 

Here, we discuss the pathologies caused by progerin and its possible link to aging in normal individuals without mutations in the *LMNA* gene. Considerable evidence reveals that a reduction in progerin in cells leads to a reduction in pathology. In the following two sections, we discuss various attempts to directly or indirectly target progerin to achieve therapeutic benefits for HGPS and address the challenges of translating them into clinical trials.

## 4. Targeting and Inhibiting Progerin at mRNA and DNA Level

Cellular progerin levels are correlated with the severity of the premature aging phenotype. Approximately 20 years ago, the specific knockdown of progerin mRNA by RNA interference was shown to alleviate cellular aging features [96,97]. Short hairpin RNA constructs were designed to target progerin mRNA and mutate pre-spliced LMNA mRNAs with 1824 C->T mutations. The expression of shRNA using lentiviruses significantly decreased progerin expression and in turn led to the improvement of the abnormal nuclear morphology, the recovery of proliferative potential, and a reduction in senescent cells [96,97]. These findings justify the potential use of gene therapy for HGPS treatment. 

Antisense oligonucleotide (ASO)-based therapies are promising strategies for the treatment of various diseases by blocking the expression of the target gene at the mRNA level through binding, followed by inactivation of the specific RNA by steric blockade or by promoting RNA degradation to specific sites on the mRNA [98,99]. The first ASO treatment in HGPS was tested in vitro by transfecting fibroblasts from patients with HGPS with an ASO targeting the progerin mRNA sequence. This approach effectively reduced progerin expression and ameliorated progerin-induced phenotypes [100]. In subsequent studies using several types of ASOs, similar results were obtained with fibroblasts from patients with HGPS [65,101,102] and from HGPS-like patients containing non-classical *LMNA* mutations that can induce progerin expression [103]. However, these studies revealed that progerin-targeting ASOs also decrease endogenous lamin A levels, raising concerns about the probable risk of lamin A depletion in vivo. However, mice lacking lamin A but maintaining lamin C expression showed no apparent phenotypes when compared to wild-type controls [65,104], encouraging researchers to test the potential of ASOs to reduce progerin levels in vivo. Another study showed that aortic progerin levels were reduced by approximately 50% in *Lmna^G609G/G609G^* mice treated with ASOs selected through an in vitro approach that targeted a 70-nucleotide region located upstream of the mutation that causes HGPS. This treatment alleviated the HGPS-associated vascular phenotype by reducing the loss of SMCs and early fibrosis; however, no longevity data have been reported [101]. Two recent studies demonstrated the promising efficacy of the ASO-dependent inhibition of progerin expression in homozygous transgenic G608G BAC mice [105,106], an HGPS mouse model that encodes human progerin and lamins A/C in addition to endogenous mouse lamins A/C [64]. The main advantage of this model over the *Lmna*^G609G/G609G^ mice is that the candidate ASOs have greater translational potential because they target the human (not mouse) *LMNA* gene. The top candidates were selected based on their ability to reduce progerin and lamin A levels without reducing lamin C expression both in vitro and in vivo. Weekly treatment of asymptomatic two-to-six-day-old to one-week-old BAC mice with selected ASOs significantly reduced progerin mRNA levels in many tissues; however, the reduction in progerin protein levels was less pronounced [105,106], similar to the results obtained in *Lmna*^G609G/G609G^ mice [65], raising concerns about the high stability of protein levels and the need for inhibitors that efficiently reduce progerin expression. In two studies, ASO treatment was shown to increase the lifespan of BAC mice by 35–60% [105,106]; however, in one study, it was shown to partially prevent SMC loss and early thickening of the ascending aorta, two key features of the HGPS vascular phenotype [106]. The use of oligonucleotides to manipulate protein production has become an important therapeutic strategy in treating genetic diseases and cancer; however, the development of chemically modified nucleic acids, bio-conjugation to escort the moiety, and the formulation of nanoparticle carriers are required for the delivery of oligonucleotide drugs [107]. Although these technological advances have led to the clinical approval of several ASO drugs, efficient and targeted delivery remains a major challenge for HGPS, in which the causative protein is expressed throughout the body and progressively affects tissues and organs.

Although ASOs have proven useful in mouse models (transgenic human G608G BAC mice) of HGPS, these models require continuous administration and do not eliminate the cause of the disease. Additionally, the treated animals die prematurely from HGPS. Considering the currently available information, adenine base editors (ABEs) appear to be more advantageous than CRISPR/Cas9 approaches because they do not induce double-stranded DNA breaks, inhibit lamin A, or efficiently correct mutations that cause HGPS. Furthermore, it prevents HGPS-associated vascular features and extends the lifespan more than any other tested treatment [108]. Recently, the transient expression of an ABE and single-guide RNA using MS2 bacteriophage-lentivirus chimeric particles corrected the mutation in 20.8–24.1% of skin cells in an HGPS mouse model (human tetop-LAG608G minigene), indicating that it could be a good approach for future gene-editing therapies [109]. However, two major challenges limit their application: moderate editing efficiency and the off-target mutagenesis of DNA and RNA. Further pre-clinical studies are required to improve the safety and effectiveness of ABE-based therapies by enhancing their efficiency, optimizing vectors, fine-tuning doses, and defining the optimal treatment duration to achieve the best outcomes for patients before moving to clinical trials.

## 5. Treatments and Clinical Trials for HGPS 

Several efforts have been made to efficiently treat HGPS using mouse models and cell lines from human patients. Treatment with the mTOR inhibitor rapamycin abolishes nuclear hemorrhage, delays cellular senescence, and enhances progerin degradation in HGPS cells via autophagic mechanisms [33,110]. Endogenous neuropeptide Y (NPY) increases caloric-restriction-induced autophagy in the hypothalamus, suggesting that NPY alleviates some features of cellular senescence in HGPS cells [111]. To restore impaired proteostasis in HGPS, the cells were treated with sulforaphane (SFN), an antioxidant derived from cruciferous vegetables, which increases the degradation of progerin by enhancing autophagic activity and reversing the premature aging features of HGPS cells [112,113]. The balance between type A lamins has been reported to be regulated by the RNA-binding protein SRSF1; therefore, one group hypothesized that the inhibition of this protein could have a therapeutic effect on HGPS and evaluated the antidiabetic drug metformin to propose a new approach to treat HGPS that could be added to the therapies currently analyzed [114]. The proteasome inhibitor MG132 was found to induce progerin clearance in classic HGPS by activating autophagy and regulating splicing [115]. Additionally, it was able to induce aberrant prelamin A clearance and improve cellular phenotypes in HGPS-like patient cells beyond those previously described in classic HGPS, providing pre-clinical evidence for a potential treatment for children with HGPS-like or classic HGPS using a promising class of molecules [116]. Recently, Zhang and colleagues found that BUBR1, a core component of the spindle assembly checkpoint, was suppressed during HGPS cellular senescence, and the remaining BUBR1 was engaged in the nuclear membrane by binding to the C-terminus of progerin, thereby limiting the function of BUBR1. Based on this, they created a unique progerin C-terminal peptide (UPCP) that efficiently blocked the binding of progerin to BUBR1 and interfered with the interaction of PTBP1 with progerin to promote the expression of BUBR1. UPCP significantly inhibited HGPS cell senescence and improved the progeroid phenotype in an HGPS mouse model (*Lmna^G609G/G609G^*) [117]. 

Several studies have reported the role of inflammatory molecules in progeria and the efficacy of therapies aimed at counteracting the pro-inflammatory state of HGPS [92,93,118]. Treatment of HGPS fibroblasts with MnTBAP/baricitinib (bar) combination therapy sustained the positive effects of bar, enhancing mitochondrial function and decreasing the levels of progerin and inflammatory factors (superoxide dismutase mimetic, MnTBAP, and JAK1/2 inhibitor, bar). Overall, co-treatment with MnTBAP/bar alleviated the abnormal phenotype of HGPS fibroblasts, making it a promising therapeutic strategy for patients with HGPS [118]. HGPS cells exhibited enhanced nuclear protein export activity due to the progerin-driven overexpression of chromosomal region maintenance 1 (CRM1). Pharmacologically inhibiting CRM1 with Leptomycin B mitigates the senescent phenotype of HGPS cells [119]. High levels of interleukin-6 (IL-6), a pro-inflammatory cytokine associated with age-related processes, have been observed in HGPS cells and mouse models (*Lmna^G609G/G609G^*). The blockade of IL-6 activity by tocilizumab, a specific antibody against the IL-6 receptor, reversed premature aging features in both HGPS fibroblasts and model mice [120]. Pharmacological inhibition of the NLRP3 inflammasome by the selective inhibitor MCC950 led to an improvement in the cellular phenotype, a significant extension of lifespan in a mouse model (Zmpste24^−/−^), and a reduction in inflammasome-dependent inflammation [95]. MG132 was able to reduce the TNF-α-induced inflammatory cytokine secretion of IL-1β, Il-6, TNF-α, IFN-γ, and TGF-β in HGPS-like patient cells [116]. Several in vitro and in vivo studies have been conducted to ameliorate bone and adipose tissue conditions in progeria. A mouse model (transgenic mice, tetop-LAG608G+; Sp7-tTA+) with the osteoblast-and osteocyte-inducible expression of progerin [121] was used to investigate the recovery from HGPS bone abnormalities by silencing the mutation and the beneficial effect of treatment with resveratrol. Complete silencing of the transgenic progerin expression normalized the bone morphology and mineralization, including improvements in the frequency of rib fractures and callus formation, an increased number of osteocytes, and normalized dentinogenesis. However, despite these positive findings, resveratrol treatment showed no beneficial effects [122]. Using an HGPS mouse model (transgenic G608G BAC), Cubria et al. (2020) showed that treatment with pravastatin and zoledronic acid significantly improved bone structure, mechanical properties, and cartilage structure parameters, thereby improving the musculoskeletal phenotype of the disease [89]. Progerin accumulation and high paracrine activation in adipocyte tissue caused chronic inflammation and cellular senescence in a *tetop-LA^G608G+^* mouse model. The pro-inflammatory cytokines IL-1α, IL-1β, IL-6, IL-17α, interferon gamma (IFNγ), and TNF-α were significantly higher than in controls [123]. Additionally, the loss of fat and fat deposits has been observed in *Lmna*^G609G/G609G^ mice [65,124]. Hartinger et al. [125] tested the effects of bar (a JAK1/2 inhibitor and an anti-inflammatory agent) and a combination of bar and lonafarnib (a farnesyltransferase inhibitor) on adipogenesis using skin-derived precursors (SKPs). Compared to mock-treated HGPS SKPs, bar and the combination of bar and lonafarnib treatments improved the differentiation of HGPS SKPs into adipocytes and lipid droplet formation, demonstrating the beneficial effect of combination treatment on adipogenesis in HGPS and other lipodystrophies [125]. Recently, HGPS-associated vascular pathological features were recovered by CRISPR/dCas9-activated Oct4 expression, which extended the lifespan of a mouse model (transgenic G608G BAC) [126]. The miR-59 was markedly upregulated in HGPS patient cells and multiple tissues in an HGPS mouse model (*Lmna^G609G/G609G^*). Treatment with AAV9-mediated anti-miR-59 reduced fibrosis in several organisms, alleviated epidermal thinning and dermal fat loss, and extended the longevity of mouse models [53]. The efforts discussed in this section for efficient HGPS treatment are summarized in Table 1.

**Table 1 cells-12-02299-t001:** The list of the various treatments for HGPS.

Treatments	Targets	Action	Study Design (Cell or Animal Model)	Refs
Rapamycin	mTOR	Autophagic induction	HGPS fibroblasts	[33,110]
Sulforaphane (SFN)	Free radicals	SFN (antioxidant derived from cruciferous vegetables) stimulates proteasome activity and autophagy in normal and HGPS fibroblast cultures	HGPS fibroblasts, Human nucleus pulposus cells, *Lmna*^G609G/G609G^ mice	[112,113]
Metformin	serine/arginine-rich splicing factor 1 (SRSF1)	A-type lamins is controlled by SRSF1. SRSF1 expression is transcriptionally regulated by the antidiabetic drug metformin.	HGPS mesenchymal stem cells, HGPS fibroblasts, *Lmna*^G609G/G609G^ fibroblasts	[114]
Pravastatin and Zoledronic acid	hydroxymethylglutaryl-CoA (HMG-CoA) reductase /bone resorption	Zoledronic acid prevents bone fractures Pravastatin is used for treating high cholesterol and preventing heart attacks and strokes	Transgenic G608G BAC mice	[89]
Neuropeptide Y	Autophagy	NPY mediates caloric restriction-induced autophagy	HGPS fibroblasts	[111]
MG132	Proteasome 26S	Autophagic activation after the loss of proteasomal activity. Downregulation of SRSF1 and reduction of inflammatory cytokines	HGPS fibroblasts, *Lmna*^G609G/G609G^ mouse, HGPS-like fibroblasts, MAD-B syndrome fibroblasts	[115,116]
Unique Progerin C-terminal peptide (UPCP)	Progerin-BUBR1	UPCP blocks the binding between Progerin and BUBR1	HGPS fibroblasts, *Lmna*^G609G/G609G^ mouse	[117]
Lonafarnib	Farnesyltransferase	Non-farnesylated prelamin A production by inhibition of farnesyltransferase activity	HGPS fibroblast, *Zmpste24*^−/−^ fibroblast, Transgenic G608G BAC mice *Zmpste24*^−/−^ mice	[119,127,128,129]
Progerinin (SLC-D011)	Progerin	The small molecule specifically bindsto Progerin and induces its degradation	HGPS fibroblasts, WRN fibroblasts and cardiomyocytes, *LMNA*^G609G/G609G^ mice	[59,96,124]
MnTBAP and Baricitinib	Peroxynitrite /Janus kinase	Inhibition of peroxynitrite- induced oxidative reactions /anti-inflammatory action by JAK1/2 inhibitor	HGPS fibroblasts	[118]
Tocilizumab	IL-6	Immunosuppression by anti-IL-6 antibody	HGPS fibroblasts, *Lmna*^G609G/G609G^ mouse	[120]
Baricitinib and lonafarnib	Janus kinase /Farnesyltransferase	Inhibition of peroxynitrite-induced oxidative reactions /anti-inflammatory action by JAK1/2 inhibitor	HGPS fibroblasts, FPLD2 syndrome fibroblasts, MAD-B syndrome fibroblasts	[125]
Anti-miR-59	microRNA-59	AAV9-mediated miR-59 inhibition by antisense oligonucleotide	HGPS fibroblasts, *Lmna*^G609G/G609G^ mouse	[53]
MCC950	NLRP3	Inflammasome inactivation by NLRP3 inhibition	HGPS fibroblast, *Zmpste24*^−/−^ mice, *LMNA*^G609G/G609G^ mice	[95]

In addition to these experimental studies on the treatment of HGPS, several clinical trials have been conducted (Table 2). Eiger BioPharmaceuticals developed orally active lonafarnib (Zokinvy™), a farnesyltransferase inhibitor, under a license from Merck & Co. [130]. The drug was initially discovered by Merck & Co. as an investigational drug for cancer; however, its development was discontinued owing to its lack of efficacy [130]. In HGPS cells, the drug inhibits farnesyltransferase activity and blocks the subsequent aggregation of progerin and progerin-like proteins in the nucleus and cellular cytoskeleton [127,131]. However, in in vitro experiments, lonafarnib showed cytotoxic effects, leading to the formation of donut-shaped nuclei [132] and cell death [133,134]. After 20 years of basic and clinical research and multiple clinical trials [128,129,135], the FDA recently approved lonafarnib (20 November 2020, in the USA) as the first drug to treat HGPS (NCT00425607). This represents an important milestone because children with progeria can be treated with medications that improve their vascular phenotypes and extend their life expectancy. However, much work remains to be done to improve the quality of life, increase life expectancy, and ultimately cure patients with HGPS. 

After the identification of the effects of zoledronate and pravastatin, a second clinical trial was initiated using these two drugs (NCT00731016), followed by a tri-therapy clinical trial combining lonafarnib, zoledronate, and pravastatin (NCT00879034 and NCT00916747). Despite the lack of additional improvements in the tri-therapy compared to lonafarnib alone, as reported in the results of the last clinical trial [136], researchers continued their efforts to find the optimal conditions for the tri-therapy combination.

Targeting the farnesylation and methylation of progerin with chemical inhibitors ameliorates some progerin-induced changes, but several questions persist concerning the underlying mechanisms. All tested strategies aimed at inhibiting isoprenylcysteine carboxyl-methyltransferase (ICMT) improved progeroid features. However, genetic inactivation and treatment with a target inhibitor (C75) appeared to increase progerin levels in cells [137]. The main disadvantage of blocking FTase and ICMT to treat HGPS is that these endogenous enzymes target several other proteins in addition to progerin. Therefore, changes in homeostatic farnesylation and methylation following FTase and ICMT inhibition may have deleterious side effects that partially offset the positive outcome of reducing the progerin-induced disease. The impact of mTOR signaling downstream of AKT should also be assessed, as mTOR haploinsufficiency extends the lifespan of the HGPS mouse model (transgenic G608G BAC) [138]. Lonafarnib and everolimus reduced pathology in an iPSC-derived tissue-engineered blood vessel model [139]. Furthermore, the combination of ICMT-targeting drugs with mTOR inhibitors such as everolimus, which are currently being tested in HGPS trials alongside lonafarnib (NCT02579044), has the potential to yield further positive effects. 

Park et al. developed treatment strategies that targeted progerin more specifically using progerinin (or SLC-D011), an inhibitor drug that directly targets progerin, induces its degradation, and finally disrupts the interaction between progerin and lamin A. The oral administration of progerinin to an HGPS mouse model (*LmnaG*^*609G*/*G609G*^) increased its lifespan by approximately 50% [59,96,124]. Before studies on diseased states, randomized, double-blind, placebo-controlled, single ascending dose (SAD) studies including food interactions were conducted. This was followed by multiple ascending dose (MAD) studies to evaluate the safety, tolerability, pharmacokinetics, and pharmacodynamic profiles of progerinin (SLC-D011) in healthy volunteers (NCT04512963). To the best of our knowledge, this is the first in-human study of progerinin. Additionally, progerinin had an excellent safety profile across all tested doses or food conditions (up to a maximum dose of 2400 mg). The study subjects in the phase I trial tolerated the drug well and confirmed an increase in exposure to progerinin under different food conditions, with fasting showing the least exposure, followed by low-fat and high-fat. The final enrollment included 63 healthy volunteers, with 47 subjects in the SAD phase and 16 subjects in the MAD phase, at one site in the USA. The drug–drug interaction potential was not clinically significant; therefore, dose adjustment was not necessary because a phase I, open-labeled, fixed-sequence study was safely completed to assess the effects of a CYP3A4 inhibitor (itraconazole) and a CYP3A4 inducer (phenytoin ER) on the single-dose pharmacokinetics of progerinin in healthy volunteers. 

## 6. Concluding Remarks

In this review, we introduce progerin as a pathogenic factor that induces HGPS (Figure 1) and provide relevant evidence in support of this. Multiple efforts to understand the causes of HGPS and develop treatments and clinical studies have been described; however, not all researchers’ contributions were included in this short review (Table 1 and Table 2). We might suggest that genome editing is the best option for the treatment of monogenic diseases. However, for systemic diseases such as HGPS, there are at least two requirements: (i) the delivery of the system to every cell in the body, and (ii) the demonstration of the absence of off-target effects in all cells. Furthermore, there is an indispensable need to strengthen the regulation of genotoxicity with regard to gene therapy in pediatric diseases. Finally, we believe that it is appropriate to expedite efficacy-based approvals for patients suffering from HGPS, as long as no toxicity risks are raised. We look forward to the upcoming clinical trials for HGPS and progeroid laminopathies and hope to learn more about the relationship between progerin and aging. 

## Figures and Tables

**Figure 1 cells-12-02299-f001:**
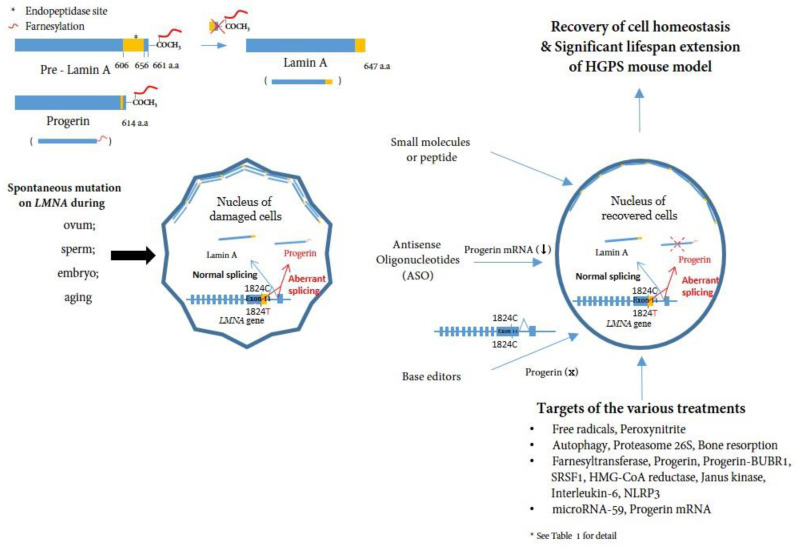
**A schematic representation of the post-translational processing of lamin A and progerin and the effect of progerin inhibition on cells.** Spontaneous mutations in LMNA (c. 1824C>T) in eggs, sperm, embryos, or during aging cause alternative splicing of the LMNA gene, leading to the accumulation of progerin in the nuclear layer. Consequently, the accumulation of progerin renders the cell unhealthy in a mechanophysiological manner. Inhibiting progerin by several methods (small molecules, ASOs, base editors) can restore damaged cells (See Table 1 for detail).

**Table 2 cells-12-02299-t002:** The list of the clinical trials for HGPS and related progeria up to date.

Clinical Trials NCT#	Drugs	Stage	Number of Individuals	Study Type	First Posted (Year) /Recruiting Status	Status or Main Finding
NCT00425607	Lonafarnib	Phase II	29	Interventional	2007 /complete	Found life span extension (about 1.6 years)
NCT00731016	Zoledronate and pravastatin	Phase II	15	Interventional	2008 /complete	Found the reduction of the alternative prenylation induced by Lonafarnib
NCT00879034	Zoledronate, pravastatin, and Lonafarnib	Phase II	5	Interventional	2009 /complete	No additional improvement of the tri-therapy as compared to lonafarnib alone
NCT00916747	Zoledronate, pravastatin, and Lonafarnib	Phase II	85	Interventional	2009 /active	Optimizing the efficient tri-therapy combination
NCT02579044	Everolimus and Lonafarnib	Phase I/II	80	Interventional	2015 /Enrolling by invitation	Defining Maximum-tolerated dose (MTD) of everolimus & efficacy of combination
NCT04512963	Progerinin	Phase I	64	Interventional	2020 /complete	No safety concerns with the Progerinin at all testing doses and food conditions (up to the maximum dose of 2400 mg)

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
