# Peer review of "Progerin, an Aberrant Spliced Form of Lamin A, Is a Potential Therapeutic Target for HGPS"

_cells, 2023, doi:10.3390/cells12182299_

Round 1

Reviewer 1 Report (Previous Reviewer 1)

This is a very deep and interesting review on progeria.  It was probably a hard task to write it. I do not think that many other reviews have been published with as much details on this disease and its possible therapies. It will be useful for scientists and medical doctors,  as well as for patients. Some paragraphs are perhaps overly detailed but this can be useful for some of the readers.

English has been seriously improved, just a few sentences still have to be improved.

I recommend this review for publication in Cells.

A few corrections have to be done, english has been strongly improved.

Author Response

Response to Reviewer 1 Comments

This is a very deep and interesting review on progeria. It was probably a hard task to write it. I do not think that many other reviews have been published with as much details on this disease and its possible therapies. It will be useful for scientists and medical doctors, as well as for patients. Some paragraphs are perhaps overly detailed but this can be useful for some of the readers.

English has been seriously improved, just a few sentences still have to be improved.

I recommend this review for publication in Cells.

  • We appreciate your comments and time for our manuscript.

Reviewer 2 Report (Previous Reviewer 2)

English is  improved and the manuscript is easier to read compared to the previous version. Manuscript is still a little bit confused in some parts. Sometimes authors begin to discuss something then they stop and they start to discuss something else and then they come back to the topic they discussed before. The paragraph “Increased Malfunction in the Phenotype by Intracellular Progerin Expression” is too long, I suggest to split it in two parts: one focused on the effect of  progerin at the cellular level (chromatin, ER, miRNA etc..) and another one in which authors can explain the main clinical effect of progerin accumulation ( systemic effect of progerin/ effect of progerin on tissues). I highlighted in green parts to modify/moove/correct.

Author Response

Reviewer 3 Report (Previous Reviewer 3)

The manuscript is now ready for publication once the authors have addressed the issues indicated by the language reviewers.  

Author Response

Response to Reviewer 3 Comments

The manuscript is now ready for publication once the authors have addressed the issues indicated by the language reviewers.  

We appreciate your comments and time for our manuscript.

Reviewer 4 Report (New Reviewer)

The review by Bae-Hoon e al, presents an vision of the Hutchinson-Gilford progeria syndrome (HGPS). The review is interesting well written and enyoyable. HGPS is a very rare disease characterized by premature and accelerating aging. The study of HGPS has attracted great interest because understanding the mechanisms underlying it will serve for intervening natural ageing.

The review is divided logically, t describes the historic part of HGPS, since the identification of the causative mutation until the molecular basis resulting in accelerate aging. In the following parts, the revie summarizes the deleterious effects of progeria at cellular and systemic levels with emphasis in cardiovascular alterations, pro-inflammatory features, oxidative stress, and the enigmatic observation that children suffering HGPS do not develop neurological disease. Finally, the review discusses the current developing therapies aimed at alleviating HGPS using cellular and mice models and the few attempts in the clinics.   

I consider the review is timely and deserves publishing in Cells in its current status

Minor points

-This sentence in line 46 seems wrong “…..prelamin A prenyl protein-specific methyltransferase”.

-Considering includes this recent paper regarding a treatment to improve mitochondrial function and oxidative stress in HGPS fibroblasts.  Monterrubio-Ledezma, et al. Cells. 2023: 2(2):275.

Author Response

Response to Reviewer 4 Comments

The review by Bae-Hoon e al, presents an vision of the Hutchinson-Gilford progeria syndrome (HGPS). The review is interesting well written and enyoyable. HGPS is a very rare disease characterized by premature and accelerating aging. The study of HGPS has attracted great interest because understanding the mechanisms underlying it will serve for intervening natural ageing.

The review is divided logically, t describes the historic part of HGPS, since the identification of the causative mutation until the molecular basis resulting in accelerate aging. In the following parts, the revie summarizes the deleterious effects of progeria at cellular and systemic levels with emphasis in cardiovascular alterations, pro-inflammatory features, oxidative stress, and the enigmatic observation that children suffering HGPS do not develop neurological disease. Finally, the review discusses the current developing therapies aimed at alleviating HGPS using cellular and mice models and the few attempts in the clinics.   

I consider the review is timely and deserves publishing in Cells in its current status

Minor points

-This sentence in line 46 seems wrong “…..prelamin A prenyl protein-specific methyltransferase”.

-Considering includes this recent paper regarding a treatment to improve mitochondrial function and oxidative stress in HGPS fibroblasts.  Monterrubio-Ledezma, et al. Cells. 2023: 2(2):275.

  • We appreciate your comments and time for our manuscript. Please look at the small changes, which was recommended by Reviewer 2. The changed parts were highlighted in yellow background and red letters.
  • “prelamin A prenyl protein-specific methyltransferase” was changed to “Third, the newly exposed farnesyl cysteine is carboxyl-methylated by a prenyl pro-tein-specific methyltransferase.
  • “Monterrubio-Ledezma, et al.  2023: 2(2):275” was added to reference list as 139.

In Line 388~391, sentences highlighted in blue sky were added as follow: “HGPS cells exhibited an enhanced nuclear protein export activity due to the progerin-driven overexpression of chromosomal region maintenance 1 (CRM1). Pharmacologically inhibiting CRM1 with Leptomycin B, mitigates the senescent phenotype of HGPS cells [139].”

  • Please look into our manuscript.

This manuscript is a resubmission of an earlier submission. The following is a list of the peer review reports and author responses from that submission.

Round 1

Reviewer 1 Report

The authors did not take into consideration a recent review in the same field published in Cells: The Molecular and Cellular basis of Hutchinson-Gilford Progeria Syndrome and potential treatment. Batista NJ et al. Genes 2023 Feb 27, 14 (3), 602. Unfortunately, they probably wrote their review before the publication of this review.

This very clear published review by Batista et al. is dealing both with the mechanisms of HSPG and possible therapies, so that the present manuscript is redundant with this published review. Too much redundant to be published in its present state and this much as in addition, it is less well written and illustrated than the published one.

To my point of view the author of the submitted review seems to have a solid expertise in the field of therapy. They should take advantage of this knowledge in order to try to propose a review complementary to the published one.

I understand that one cannot explain therapies without background on the disease. However, the authors may re-built the review by creating different sections in which they are focusing on one HSPG mechanistic aspect in direct link with a possible therapy that they will describe in the given section. By the way, they may strongly reinforce the description of the alternative possible therapies and their expected chances of success.

Another remark concerning therapies: there is no clear mention of possible immunotherapies.

If such modifications were not possible, the review would be considered as too much redundant with the recent one, less well written and less well illustrated to be published in Cells.

The present figure is indeed extremely poor.

The authors have provided a large series of references. However, some recent ones especially in link to therapies would deserve to be mentioned and discussed:

Lonafarnib and everolimus reduce paththology in iPSC-derived tissue engineered blood vessel model of HSPG Sci Rep 2023 13, 5032

Transcriptional activation of endogeneous Oct4 via CRISPR/dCas9 activator accelerate HSPG…. Aging Cell, 2023, e13825.

Plasma Progerin in patients with HSPG immune assay development and clinical evaluation Gordon LB et al. Circulation 2023

Anti-has-miR-59 alleviates premature senescence associated with HSPG in mice. Hu Q et al. EMBO J 2023 42(1) e110937

Impact of MnTBAP and Baricitinib treatment on HSPG fibroblast. Vehns E et al. Pharmaceuticals (Basel) 2022, 15, 945

Transient expression of an adenine base editor corrects HGPS mutation and improves the skin phenotype in mice. Whisenant D et al. Nat Commun 2022, 13

MiR-376a-3p and miR-376b-3p overexpression in HSPG fibroblast inhibits cell proliferation and induces premature senescence Frankel et al. iScience 2022, 25, 103757.

In addition to these scientific points, English has to be strongly revised.

Examples:

Lane 116: From this review, we going to                      are is missing after we

Lane 195: In this aspect, several                             respect instead of aspect

Lane 211: people realize that                                it was found that

Lane 228: similar results came in fibroblast    were obtained with fibroblast

Lane 236: a region 70 nts upstream                 located 70 nts upstream

Lane 530: was it shown                                     it was shown 

English has to be seriously improved as indicated in the  above text for the authors.

Reviewer 2 Report

The present review gives a general overview of the effect of progerin on various tissues and the possible therapeutic approaches developed to solve problems due to progerin accumulation. I found the review a little bit superficial. The role of inflammatory molecules in Progeria and the efficacy of therapies aimed at counteracting the proinflammatory state in HGPS are not even mentioned. In addition, there are little information on bone and fat tissues disorders associated with progerin accumulation and on the attemptative strategies developed to solve these problems. I suggest to mention these studies.

Moreover there are many conceptual mistakes, first of all the wrong information reported several times in the text regarding the diagnosis of Progeria. The Progeria is diagnosed within the first 6 months of age. This wrong information could be misleading for readers not involved in the study of Progeria.

I suggest to explain better the results obtained by other authors. For example: the sentence “In response to DNA damage, it has been proposed that lamin A/C is a substrate for nucleophagy that leak nuclear DNA through interaction with LC3” does not recapitulate well results obtained by Li et al.

Please check references (ref 44 is Mateos not Marìa; insert reference Dreesen O, Chojnowski A, Ong PF, Zhao TY, Common JE, Lunny D, Lane EB, Lee SJ, Vardy LA, Stewart CL, Colman A. Lamin B1 fluctuations have differential effects on cellular proliferation and senescence. J Cell Biol. 2013 Mar 4;200(5):605-17. doi: 10.1083/jcb.201206121. Epub 2013 Feb 25. PMID: 23439683; PMCID: PMC3587829.) 

Sometimes it is difficult to follow what authors want to tell because of the poor English. There are many sentences without verb and others difficult to understand. For this reason I suggest an extensive editing of English language.

Reviewer 3 Report

This is a review article intended for the Special Issue “Advances in Treatments for Hutchinson-Gilford Progeria Syndrome”. The topic is of utmost interest, as only recently a drug has been made available for the treatment of this devastating disease that, in turn, constitutes a model of accelerating ageing. However, the manuscript raises major concerns:

1. The manuscript phrasing and grammar should be carefully revised throughout the text. The language is so messy that the text is hard to follow. To name just a few examples: a. The title is meaningless. This reviewer can come to interpret that the title tries to refer to progerin, the HGPS-inducing protein, and to its potential as an HGPS therapeutic target. Please, use an appropriate title aligned with the manuscript scope; b. Line 67: What does “Another way” mean here?; c. Line 143: “Marí a and her colleague” should read “Mateos et al.”; d. Lines 112 and 163: Allusion to a “culprit protein” is unnecessary; e. Line 167: “Dressen and his colleague” should read “Chojnowski et al.”; f. Line 211: What does “people realize” mean here?

2. From lines 16-21 (which should be, however, rephrased), the abstract appears to indicate that the manuscript reviews i) the discovery of progerin as the HGPS causative agent; ii) the evidences showing the detrimental effects of progerin as well as the beneficial effects of its suppression; and iii) the clinical trials on HGPS treatment. Please rewrite to reflect the manuscript scope and contents.

3. Introduction: shorten the current version and include an introduction to Sections 2, 3, 4, and 5.

4. Figure 1 should be improved regarding both design and contents. While a significant part of the figure area is blank, several words and shapes cannot be read due to their reduced size. In addition, the caption is too short and does not describe the process that is schematized in the figure. ASO is not defined until Section 4.

5. Section 3: While this Section, which is aligned with point ii) in comment #2 above, includes a significant number of “old” references, some relevant works (e.g. at the protein or RNA level) are missing here that would strengthen the evidences presented. In addition, I am missing here comments on i) the several progeroid animal models developed to date; and ii) the evidences that show a correlation between progerin level and HGPS severity, how this level can be measured at the protein and RNA levels in cells and tissues from HGPS patients or animal models.

6. Section 5: Reorganize to include all relevant clinical trials (only three of them are mentioned). A table listing their main features (phases, number of individuals, study type, main findings, etc.) would be useful.

7. Section 6: Avoid unnecessary generalities (lines 358-364). Regrettably, gene therapy is not at reach, please broaden your conclusions

The manuscript phrasing and grammar should be carefully revised throughout the text. The language is so messy that the text is hard to follow

Round 2

Reviewer 1 Report

The authors followed recommendations of the reviewers, organization of the manuscript has been seriously improved. In its present form the manuscript is quite less redundant with previous reviews dealing with HGPS and progerin.

However, I consider that the unique Figure in the manuscript would still deserve serious improvements. It is quite small and is not very informative. 

The Table on the clinical status of treatments is fine. However, as the manuscript is dealing with various putative treatments for HGPS some having just been tested in animals, the authors may in addition design a panel in the present Figure or built an additional Figure summarizing the various targets which have been tested.

Furthermore, they should add a Table listing the various treatments mentioned in the text, with the methods used for the administration, whether they were performed on animal or human, the results obtained, the reference numbers. This would help the readers to have a rapid view of the various treatments already tested on animals and on humans.

Finally, English writing although better compared to the first version of the manuscript, still has to be seriously improved in some parts of the manuscript.

With these improvements the manuscript can be published.

English has been improved but there are still several sentences to be corrected. This is a need before publication.

Reviewer 2 Report

I appreciated that Authors followed my suggestions reporting some information on adipose tissue and bone alteration in Progeria but in my opinion this part is still poor. I suggest to improve this part by reporting studies made both in vitro and in vivo in order to ameliorate bone and adipose tissue condition (kyphosis, osteolysis, high fat diet, treatments..). Moreover, I would suggest to clarify the role of inflammatory molecules in Progeria instead of writing The role of inflammatory molecules in Progeria and the efficacy of therapies aimed at counteracting the proinflammatory state in HGPS have been reported.

I’m quite surprise to still find sentences such as “HGPS could not be diagnosed when the patient was born, but after the age of 2 years, visible, evident symptoms can be observed.” Please reformulate these sentences considering that Progeria is diagnosed within the first 6 months of age.

Although the manuscript has undergone English correction, many mistakes are still present and the manuscript results still difficult to follow in some parts. Authors should pay attention not only to the grammar, but also to the meaning of sentences. Please avoid to use complicate roundabout expressions. I suggest to use short and simple sentences. For example in the abstract instead of writing “Furthermore, it will be considered a moment to broaden the horizon on the cause of human aging” authors could write something like “studying progeria could be useful to unravel the cause of human aging” .

 I saw that authors added animal models used to study progeria, but they just made a list. I suggest to write something more, explaining the utility of these models (it was generate to study what?). For example authors can insert the specific animal model used to test the effect of Tocilizumab or Pravastatin plus Zoledronic Acid instead to simply write “HGPS model”. Another possibility is to dedicate a paragraph on animal models and their utility.

Other considerations:

Please rewrite the abstract since it does not recapitulate well the manuscript.

I suggest to split the first paragraph in two: one more general (overview on Progeria clinical aspects/mutations and description of lamin A/C processing) and a second paragraph focused on neurological involvement in Progeria.

Please pay more attention to the text: avoid truncated words (for example HGP instead of HGPS), avoid to be repetitive or redundant; when you mention a paper, remember to report the name of the first author (ref 42) otherwise you can write the group ofgroup leader name.

Please substitute “the culprit protein” with “Progerin”, you have already pointed out that progerin is the cause of progeria at the beginning of your manuscript.  

Please use italics for the name of genes. 

Although the manuscript has undergone English correction, many mistakes are still present and the manuscript results still difficult to follow in some parts. Please pay attention not only to the grammar, but also to the meaning of sentences.

Reviewer 3 Report

Major concerns:

1.     In the previous review, I raised the following concern:

The manuscript phrasing and grammar should be carefully revised throughout the text. The language is so messy that the text is hard to follow. To name just a few examples:

a.      The title is meaningless. This reviewer can come to interpret that the title tries to refer to progerin, the HGPS-inducing protein, and to its potential as an HGPS therapeutic target. Please, use an appropriate title aligned with the manuscript scope;

b.     Line 67: What does “Another way” mean here?;

c.      Line 143: “María and her colleague” should read “Mateos et al.”;

d.     Lines 112 and 163: Allusion to a “culprit protein” is unnecessary;

e.      Line 167: “Dressen and his colleague” should read “Chojnowski et al.”;

f.        Line 211: What does “people realize” mean here?

As stated above, these were “just a few examples”, which in some cases have been taken care of inadequately (e.g. “As shown in this review, it can be realized that the level of Progerin in the cell…” should probably be “The level of cellular progerin correlates with the severity of the premature aging phenotype”) and in some other cases have just been ignored, as shown below. The manuscript still lacks a careful revision regarding phrasing and grammar. Just a few additional random examples (I regret that the line numbering is missing in the revised version of the manuscript):

·        Abstract: “HGPS is an ultra-rare disease indicating that the aging …”

·        Abstract: “Through this disease, it was recognized that human aging in old people…”

·        Page 1, last line: “… through four-step modifications…”. Do the authors mean “through four sequential posttranslational processing steps?”

·        Page 2, 1st paragraph: “… the endopeptidase removes terminal 15 amino acids…”. Do the author mean “the endopeptidase removes the carboxyl-terminal 15 amino acids”

·        Page 2, 2nd paragraph: “… from HGP patients…”

·        Page 2, 2nd paragraph: “… generates abnormal variant protein, called as Progerin…”

·        Page 2, 3rd paragraph: The meaning of “Laminopathies” has already been explained a few lines above.

·        Page 2, 3rd paragraph: “that the patients” should read “that HGPS patients”.

·        Page 2, 3rd paragraph: “… but the brain generates mostly lamin C but very little lamin A…”

·        Page 2, 3rd paragraph: “… glial cells in the brain, whereas there is little lamin A expression in the brain…”

·        Page 2, 4th paragraph: “Most scientists have speculated…”

·        Page 2, 4th paragraph: “… it is not about…”

·        Page 2, last line and page 3: “In this point, this phenomenon for the regulation of lamin A in the brain gives us a chance to think about Progerin, non-properly processed and toxic lamin A : children with HGPS have aging-like phenotypes in many tissues but are absent of common features of physiologic aging in the CNS such as senile dementia. Therefore, the Progerin level in cells could be a pathology-inducing factor because HGPS patients have Progerin accumu-lation in most tissues, but not in neurological tissues” What is the point here?

·        Page 4, first 3 lines: References 30 and 31 are used here regarding the reduced expression levels of Progerin and lamin A and C (lamin A/C) in induced pluripotent stem cells (iPSCs) derived from HGPS patients, while the third paragraph of page 2 resorts to references 20-23 for the same evidence… Is this alright?

·        Page 4, 1st paragraph: “… the patient’s HGPS cells…” Are we actually dealing here with the HGPS cells from a single patient?

·        Page 4, 1st paragraph: “This may explain the reason HGPS could not be diagnosed when the patient was born, but after the age of 2 years, visible, evident symptoms can be observed” Which patient is this? Why are mixed verb tenses used?

·        The “culprit protein”, which the authors claimed to have removed, is still in the manuscript (pages 4 and 5).

·        Page 4, 1st paragraph: “From this review” is incorrect. The whole last sentence of the 1st paragraph on page 4 is not needed here.

·        Page 4, 2nd paragraph: “… in the nuclei of a vascular cell”. Do the authors mean “in the nucleus of vascular cells”?

·        Page 4, 2nd paragraph: “… triggering plaque vulnerability, and reduce lifespan”

·        Page 4, 2nd paragraph: Andrés “and his colleagues” did not develop the CRISPR-Cas9 technology. “His” or “her” colleagues should be avoided throughout the manuscript.

·        Page 4, 3rd paragraph: Why is the isotopic labelling procedure (iTRAQ) mentioned?

·        Page 5, 1st paragraph: Patients cannot represent skeletal dysplasia.

·        Page 5, 1st paragraph: “and short stature and HGPS”

·        Page 5, 1st paragraph: “HGPS model mouse”.…

The list is very very large. This is really surprising given that, according to the authors, the manuscript has undergone English correction.

2.     The Abstract has been modified to accommodate my suggestions, but the remainder (i.e. the first eight lines) should be fully rewritten for grammatical, phrasing and logical reasons.

3.     I definitively did not suggest in my previous revision that the “Introduction” Section had to be moved to (former) Section 2. I suggested to shorten the “Introduction” Section on the one hand and, on the other hand, to include (in the Introduction) some type of short outline of Sections 2, 3, 4, and 5 of the original manuscript.

4.     Figure 1 has been improved, but things like “Franesylation” and “model mouse” remain.

5.     In my previous review I recommended to comment in (former) Section 3 works presenting evidences about the detrimental effects of progerin as well as the beneficial effects of its suppression. The authors have added three works in the revised version, but still they are omitting relevant contributions (e.g. at the protein or RNA level).

6.     With respect to the progeroid animal models, the authors just mention them in the “Efforts to develop treatments and clinical trials for HGPS” Section of the revised manuscript, which I believe is not the optimal place. The authors should substantiate in the “Increased malfunction phenotype by intracellular progerin expression” Section how these models have contributed to HGPS research, while the last lines of the first paragraph in the “Efforts to develop treatments and clinical trials for HGPS” Section, which are meaningless, should be removed.

7.     Reference 101 of the revised manuscript is not the only paper dealing with the correlation between progerin level and HGPS severity. Additional relevant works must be considered. Of note, reference 101 of the revised manuscript is a good example of a paper presenting methodology for measuring progerin levels at the protein or RNA levels in cells and tissues from HGPS patients or animal models (blood in this case), an issue around which the authors have just ignored my recommendations.

8.     In my previous revision I recommended to avoid unnecessary generalities (former lines 358-364) in the “Concluding remarks” Section, and to broaden these conclusions (so that these are not focused on a gene therapy that is regrettably not at reach yet). The authors have just deleted former lines 358-364, but now the whole Section needs to be fully rewritten to accommodate additional conclusions other than those related to gene therapy.  

The manuscript grammar and phrasing should be carefully revised.

Round 3

Reviewer 1 Report

The authors very seriously took all recommendations into consideration and the review has been strongly improved as compared to the initial version. I think that it deserves publication with only minor English improvement. It will be useful for people interested to have an overview on the present state of international investigations on Progeria treatment.

Slight improvement needed can be done with the Cells's edition service.

Reviewer 2 Report

I really appreciated that authors followed my suggestions and I applaud their efforts, but although I found the manuscript improved compared to the original version, it still presents many problems. First, the poor attention in grammar ad phrasing revision makes the manuscript difficult to understand. Second, paragraphs are too long, making the text confusing and hard to follow. Third, there are some conceptual mistakes (e.g. the sentence “LMNA actually produces two proteins as a result of alternative splicing: pre-lamin A (the precursor protein to lamin A) and lamin C” in not correct since LMNA gene encodes four proteins: lamin A, lamin C and minor products lamin C2 and lamin A delta 10). Therefore, in its current state, the manuscript is not suitable for publication in Cells. Overall, I believe that this manuscript has a good potential but it needs a rewriting to make it easier and a thorough language editing.

The language and sentence structure of this manuscript are at times incomprehensible. The paper needs rewriting and extensive language editing. 

Reviewer 3 Report

The authors have once more limited themselves to addressing the few specific corrections that this reviewer listed as examples. Consequently, the manuscript once more lacks the careful revision of phrasing and grammar required to achieve the standard for publication.   

The manuscript once more lacks the careful revision of phrasing and grammar required to achieve the standard for publication.